# Hierarchical Decomposition Framework for Feasibility-hard Combinatorial Optimization

## Abstract

Combinatorial optimization (CO) is a widely-applied method for addressing a variety of real-world optimization problems. However, due to the NP-hard nature of these problems, complex problem-specific heuristics are often required to tackle them at real-world scales. Neural combinatorial optimization has emerged as an effective approach to tackle CO problems, but it often requires the pre-computed optimal solution or a hand-designed process to ensure the model to generate a feasible solution, which may not be available in many real-world CO problems. We propose the hierarchical combinatorial optimizer (HCO) that does not rely on such restrictive assumptions. HCO decomposes the given CO problem into multiple sub-problems on different scales with smaller search spaces, where each sub-problem can be optimized separately and their solutions can be combined to compose the entire solution. Our experiments demonstrate that this hierarchical decomposition facilitates more efficient learning and stronger generalization capabilities in terms of optimality of the solution. It outperforms traditional heuristic, mathematical optimization, and learning-based algorithms on Steiner Tree Packing Problem (STPP), a problem that cannot guarantee a feasible solution when using the hand-designed process.

## 1 Introduction

The development of efficient algorithms for solving NP-hard problems, such as combinatorial optimization (CO) problems, is of central interest to many industries. However, a significant challenge in the development of CO algorithms is the requirement for in-depth domain knowledge to manually design a heuristic algorithm specific to a given CO problem. Additionally, such algorithms often lack scalability to other CO problems. To circumvent this challenge, supervised learning (SL)-based models were proposed to solve complex CO problems (Vinyals et al., 2015; Joshi et al., 2019; Gasse et al., 2019; Paulus et al., 2022) These approaches utilize neural networks to directly predict the optimal solution given a problem instance. However, a limitation of SL-based methods is the requirement for the optimal solution as a learning target during training, as finding the optimal solution for many CO problems is computationally intractable. Additionally, the supervision provided in SL-based approaches does not take into account the quality of the solution, such as the cost objective value. The lack of consideration for solution quality can result in a situation where, despite a small error in the model's predictions, the quality of the solution may be arbitrarily poor (*i.e.*, suboptimal). Then, reinforcement learning (RL)-based approaches were proposed as an alternative, since it can directly learn from the CO objective (*i.e.*, solution cost) without requiring the optimal solution. RL-based approaches formulate the CO problem as a sequential decision making problem. At each step, the neural network predicts an action to update the previous partial solution and the reward is given based on the quality of the constructed solution. However, the main limitation of RL-based approaches is the ill-defined reward (or the solution cost) for *infeasible solutions*. This limitation has been often ignored in previous works (Bello et al., 2017; Khalil et al., 2017; Kool et al., 2019) as the feasibility of the solution is guaranteed by properly designing the action space in certain CO problems such as the traveling salesman problem (TSP). However, many real-world CO problems have complex constraints where action space design *cannot* guarantee the feasibility (*e.g.*, circuit wiring (Grötschel et al., 1997), routing problem with time windows (Ma et al., 2020)). We refer

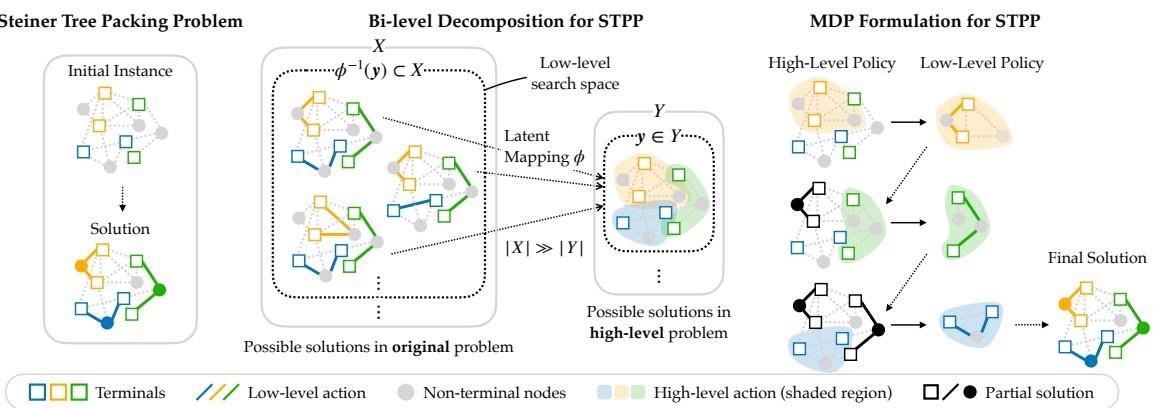

Figure 1: (Left) We tackle the Steiner tree packing problem that aims to find a minimum-weight tree spanning all the terminal nodes (square boxes) for each type (color) without overlap. (Middle) We propose to decompose the given problem into high-level and low-level sub-problems via mapping $\phi : X \mapsto Y$, and solved separately. This facilitates the learning since the search spaces are much smaller for high-level problem $|Y| \ll |X|$ and low-level problem $|\phi^{-1}(\mathbf{y})| \ll |X|$ compared to the original problem. (Right) In MDP formulation, high-level agent chooses a set of nodes (shaded region) to define a sub-graph for each terminal type (color), and low-level agent finds a tree spanning all the terminal nodes within the sub-graph.

such constraints as *feasibility-hard* constraints and the CO problem with such constraints as feasibility-hard CO problems in the rest of the paper.

A common approach to address the issue of ill-defined reward is assigning the lowest possible reward, such as 0, for an entire episode when an infeasible solution is predicted. However, this approach can lead to inefficiency and instability in RL training due to the sparse supervision, known as the sparse reward problem.

To address this issue, we propose a hierarchical decomposition framework for tackling the feasibility-hard CO problems. Specifically, we focus on the Steiner tree packing problem (STPP) that aims to find a tree spanning all the *terminal* nodes with minimum weights (see Figure 1 for illustration). The proposed framework decomposes the solution search space into high-level and low-level space via latent mapping. Intuitively, the high-level solution suggests a sub-problem (*e.g.*, a subset of variables, constraints, or a sub-graph) and the low-level solution is the solution to the suggested sub-problem. We claim that the proposed decomposition provides several advantages. First, by separating the problem into smaller sub-problems (*i.e.*, separation of concern), the proposed framework facilitates the learning. Second, the high-level agent partitions the graph such that low-level agent observes sub-problem of similar size, regardless of the input problem size. This helps the model scale to larger problems more easily, as the distribution shift is reduced. We empirically evaluate HCO against competitive baselines on large-scale STPP instances. Our results demonstrate the effectiveness of the proposed method in comparison to the baselines.

**Contribution.** We summarize our contributions:

- To our knowledge, this is the first work to solve Steiner tree packing problem utilizing an end-to-end learning framework.

- We propose a novel decomposition approach for general CO problems that results in sub-problems with smaller search spaces.

- We demonstrate that the proposed approach significantly improves the learning and generalization.

## 2 Preliminaries

### 2.1 Combinatorial Optimization as Markov Decision Process

Combinatorial optimization (CO) is a mathematical optimization over a finite set, with a discrete feasible solution space. Formally, a combinatorial optimization problem can be written as follows.

$$\arg\min_{\boldsymbol{x} \in X} \{f(\boldsymbol{x}) : \boldsymbol{x} \in \mathcal{F}\} \tag{1}$$

where $X$ is a finite support for the variable $\boldsymbol{x}$, $\mathcal{F} \subset X$ is a set of feasible solutions[1], and $f : X \to \mathbb{R}$ is an objective function of the CO problem. For instance, a mixed integer linear programming (MILP) problem with $n$ variables and $m$ constraints can be written in the form

$$\arg\min_{\boldsymbol{x} \in \mathbb{Z}^p \times \mathbb{R}^{n-p}} \{\boldsymbol{c}^\top \boldsymbol{x} : \boldsymbol{A}\boldsymbol{x} \leq \boldsymbol{b}, \ \boldsymbol{x} \geq \boldsymbol{0}\} \tag{2}$$

where $\boldsymbol{A} \in \mathbb{R}^{n \times m}$, $\boldsymbol{b} \in \mathbb{R}^m$, and $\boldsymbol{c} \in \mathbb{R}^n$.

Most of the CO problems can be formulated as a Markov Decision Process (MDP) (Khalil et al., 2017; Gasse et al., 2019). Formally, it makes two assumptions to the CO problem: 1) the solution space $X$ of the original problem (1) is a finite vector space and 2) the objective $f$ is *linear* on $X$, so that for any given decomposition of $X$ into direct sum of subspaces $X = X_1 \oplus \cdots \oplus X_n$, we have $f(\boldsymbol{x}) = \sum_{i=1}^n f(\boldsymbol{x}_i)$ for each $\boldsymbol{x}_i \in X_i$.[2] Then, the original problem (1) can be written as the following sequential decision making problem:

$$\arg\min_{\substack{\boldsymbol{x}_t \in X_t, \ \forall t=1,\cdots,H \\ X = X_1 \oplus \cdots \oplus X_H}} \{\sum_{t=1}^H f(\boldsymbol{x}_t) : \sum_{t=1}^H \boldsymbol{x}_t \in \mathcal{F}\}. \tag{3}$$

The sequential decision can be thought of choosing for each timestep $t$ an action $\boldsymbol{x}_t \in X_t$, to receive a reward $\mathcal{R}(s_t, a_t) = -f(a_t)$ and a large negative penalty $c \leq -\sup_{x \in X} f(x)$ if and only if any future choice of action inevitably leads to an infeasible solution at the end of the horizon. The optimal policy $\pi^* \in \Pi$ for the original problem can be found upon maximizing the expected return

$$\pi^* = \arg\max_{\pi \in \Pi} \mathbb{E}^\pi \Big[ \sum_{t=1}^H \mathcal{R}(s_t, a_t) \mid s_0 \Big]. \tag{4}$$

We defer the rest of the details to Appendix A.1. Note that for some CO problems (*e.g.*, TSP, MVC, Max-Cut), carefully designing the action space can make the constraint trivially satisfied (Khalil et al., 2017), where in this case, reinforcement learning algorithm can efficiently solve the problem. However, when the constraint satisfaction is not guaranteed, the reinforcement learning methods often suffer from the sparse reward problem, and does not learn efficiently. In this work, we focus on the challenging CO problems, where designing action space cannot guarantee the constraint satisfaction (*i.e.*, *feasibility-hard* constraint): the Steiner tree packing problem.

### 2.2 Steiner Tree Packing Problem

A Steiner tree problem (STP) can be thought of as a generalization of a minimum spanning tree problem, where given a weighted graph and a subset of its vertices (called terminals), one aims to find a tree (called a Steiner tree) that spans all terminals (but not necessarily all nodes) with minimum weights. Although minimum spanning tree problem can be solved within polynomial time, the Steiner tree problem itself is a NP-complete combinatorial problem Karp (1972). Formally, let $G = (V, E)$ be an undirected weighted graph, $w_e$ for $e \in E$ its edge weights, and $T \subset V$ be the terminals. Then, a Steiner tree $\mathcal{S}$ is a tree that spans $T$ such that its edge weight is minimal. Hence, the optimization problem for STP can be written as follows.

$$\arg\min_{\boldsymbol{x} \in 2^E} \{\sum_{e \in \boldsymbol{x}} w_e : \boldsymbol{x} \in \Sigma_T\} \tag{5}$$

---

[1] $\mathcal{F}$ is either discrete itself or can be reduced to a discrete set.

[2] All CO problems that admits MILP formulations, such as STPP, TSP, BPP, Max-Cut, etc, satisfy the above assumptions.

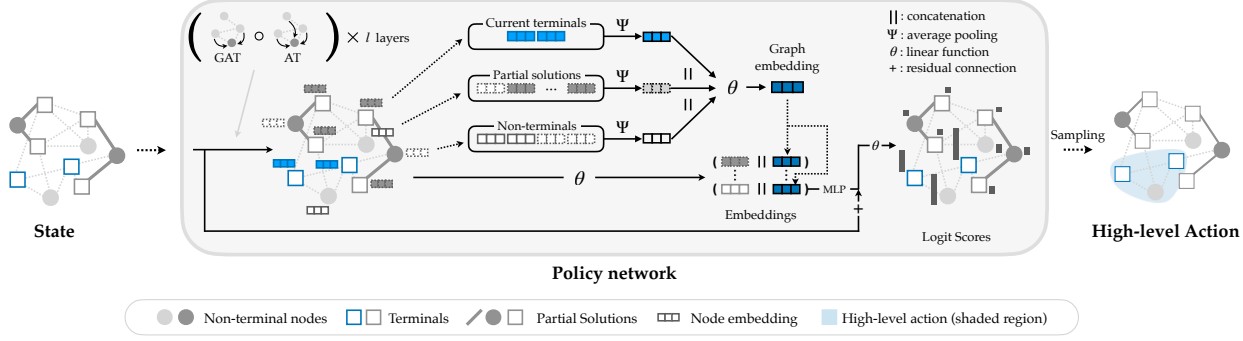

Figure 2: Illustration of the our policy model for STPP. 1) The agent observes a state consisting of a graph, terminals (rectangular nodes), and partial solution (thick gray nodes and edges). 2) The agent first extracts the node and edge features, and feed it to the GAT+attention module. 3) The GAT+attention module computes the global graph embedding by aggregating the node embeddings of each type. 4) The global graph embedding is then appended to each node feature, 5) and transformed to compute the logit score for each node. 6) Finally, the action is sampled according to the logit score.

where $2^E$ is a power set of $E$, and $\Sigma_T$ is a set of all Steiner trees that span $T$. A more generalized version of the above Steiner tree problem is called the Steiner tree packing problem, (STPP) where one has a collection $\mathcal{T}$ of $N$ disjoint non-empty sets $T_1, \cdots, T_N$ of terminals called *nets*, that has to be *packed* with disjoint Steiner trees $\mathcal{S}_1, \cdots, \mathcal{S}_N$ spanning each of the nets $T_1, \cdots, T_N$. The optimization problem for STPP can be written similarly, with $N$ variables.

$$\underset{\boldsymbol{x}_1,\cdots,\boldsymbol{x}_N \in 2^E}{\arg\min} \{ \sum_{\substack{n \leq N \\ e \in \boldsymbol{x}_n}} w_e : \boldsymbol{x}_n \in \Sigma_{T_n}, \ G[\boldsymbol{x}_n] \cap G[\boldsymbol{x}_m] \underset{\forall n,m}{=} \emptyset \} \tag{6}$$

where $G[\boldsymbol{x}]$ is a subgraph of $G$ generated by $\boldsymbol{x} \subset E$.

## 3 Bi-level Decomposition for Combinatorial Optimization

The goal of this section is to formulate our bi-level framework for a general CO problem (1) and the corresponding MDP, which allows us to use a hierarchical reinforcement learning policy that efficiently learns to solve CO problems.

Let us introduce a continuous surjective latent mapping $\phi : X \to Y$ onto a vector space $Y$ such that $|Y| \ll |X|$. Then, the problem (1) admits a *hierarchical* solution concept:

$$\boldsymbol{y}^* = \underset{\boldsymbol{y} \in Y}{\arg\min} \{ f(\mathrm{L}(\boldsymbol{y})) : \phi^{-1}(\boldsymbol{y}) \cap \mathcal{F} \neq \emptyset \} \tag{7}$$

$$\text{where,} \quad \mathrm{L}(\boldsymbol{y}) := \underset{\boldsymbol{x} \in \phi^{-1}(\boldsymbol{y})}{\arg\min} \{ f(\boldsymbol{x}) : \boldsymbol{x} \in \mathcal{F} \}. \tag{8}$$

We refer to problem (7) as a *high-level* problem, and (8) as a *low-level* sub-problem induced by the high-level action $\boldsymbol{y}$ in (7). Note that the hierarchical solution concept still attains an optimality guarantee of the original problem, since $\phi$ is a surjection and $X$ is finite.

The advantages of such hierarchical formulation are: (*i*) searching for feasible solutions over $Y$ rather than $X$ reduces the size of search space; (*ii*) learning to obtain an optimal solution can be done by two different learnable agents, (namely the high-level agent and the low-level agent for (7) and (8), respectively) where the task for each agent is reduced to be easier than the original problem, and (*iii*) the generalization capability (with respect to the problem size) increases when using learnable agents, since the high-level agent can be made to always provide sub-problems (for the low-level agent) with the same size, regardless of the size of the original problem.

### 3.1 Hierarchical Policy for Bi-level Optimization

In this section, we provide a learning framework that efficiently solves a general CO problem using our bi-level decomposition. In particular, we propose to employ a hierarchically structured policy to sequentially solve bi-level optimization problems in (7) and (8), in a similar spirit to reinforcement learning solving CO problem in Section 2.1. For any given subspace decomposition $Y = Y_1 \oplus \cdots \oplus Y_N$, we sequentially solve a low-level sub-problem from $\text{span}(\phi^{-1}(Y_j))$ for each $j \leq N$, since $\text{span}(\phi^{-1}(Y_j))$ is a subspace of $X$. Hence, we are able to formulate a high-level MDP $\mathcal{M}_{\text{hi}}$ by constructing $Y = Y_1 \oplus \cdots \oplus Y_N$ sequentially as done in Section 2.1, and the low-level MDP $\mathcal{M}_{\text{lo}}$ from Equation (8). Learning can be done upon training a policy network $\pi_\theta$ parameterized by $\theta$, using reinforcement learning (*e.g.*, policy gradient methods). Theorem 3.1 shows how our bi-level decomposition benefits the learning in terms of sample complexity. Full illustration of the bi-level optimization is provided in Appendix A.1.

**Theorem 3.1** (Sample Complexity Reduction). *Let $\mathcal{M}$ be a H-step MDP as in (3). For any algorithm $\mathcal{A}$ that has access to $\mathcal{M}$ which outputs a policy $\pi_\theta$ such that $V^{\pi_\theta}(s) \geq V^*(s) - \epsilon$ with probability greater than $1 - \delta$ for a given state $s$, $\mathcal{A}$ must make at least $\Omega(\epsilon^{-1}|X|^2 H \log(1/\delta))$ calls of $\mathcal{M}$, whereas $\Omega(\epsilon^{-1} \max(|Y|, |X|/|Y|)^2 H \log(1/\delta))$ calls of either $\mathcal{M}_{hi}$ or $\mathcal{M}_{lo}$ is sufficient when using a bi-level decomposition described as in (7) and (8).*

See Appendix A.2 for the proof of the above theorem. One added benefit of our hierarchical framework is that we can use different approach for each hierarchy: learning-based, heuristic, or mathematical optimization. Our choice of approach is covered in Section 3.3.

### 3.2 Hierarchical Decomposition for Steiner Tree Packing Problem

For CO problems defined on a weighted graph $G = (V, E)$ with edge weights $w_e$ for each $e \in E$, it is straight-forward and beneficial to choose a latent mapping $\phi : E \to V$ from the set of edges to the set of nodes.[3] Specifically, we consider a version of $\phi$ such that for any input $\boldsymbol{x} \subset E$, $u \in \phi(\boldsymbol{x})$ if and only if $(u, v) \in \boldsymbol{x}$ for some $v \in V(G)$. Such a mapping $\phi$ satisfies $\phi^{-1}(\boldsymbol{y}) = G[\boldsymbol{y}]$ for any set of vertices $\boldsymbol{y} \subset V$, where we slightly overload the notation for the generated subgraph $G[\boldsymbol{y}]$. For example, the high-level problem of a STPP (6) can be written as follows.

$$\underset{\boldsymbol{y}_1, \cdots, \boldsymbol{y}_N \in 2^V}{\arg \min} \Big\{ \sum_{\substack{n \leq N \\ e \in \mathrm{L}(\boldsymbol{y}_n)}} w_e : \mathrm{L}(\boldsymbol{y}_n) \in \Sigma_{T_n}, \ \boldsymbol{y}_n \cap \boldsymbol{y}_m \underset{\forall n, m}{=} \emptyset \Big\} \tag{9}$$

which is now a node-selection problem, (instead of the original edge selection problem) where $\mathrm{L}(\boldsymbol{y}_n)$ is a solution of the low-level subproblem:

$$\mathrm{L}(\boldsymbol{y}_n) := \underset{\boldsymbol{x} \in E(G[\boldsymbol{y}_n])}{\arg \min} \Big\{ \sum_{e \in \boldsymbol{x}} w_e : \boldsymbol{x} \in \Sigma_{T_n} \Big\} \tag{10}$$

Notice that the high-level problem (9) has a reduced size search space (from $2^E$ to $2^V \approx O(2^{\sqrt{E}})$), and the low-level subproblem corresponds to a single STP of a smaller subgraph $G[\boldsymbol{y}_n]$. Therefore, the original NP-hard problem (6) is decomposed into two smaller NP-hard problems. The overview of our hierarchical decomposition method for STPP is illustrated in Figure 1.

### 3.3 MDP Formulation and Hierarchical Policy for Steiner Tree Packing Problem

The high-level MDP $\mathcal{M}_{\text{hi}}$ for STPP is based on a sequential decision making $\boldsymbol{y}_1, \cdots, \boldsymbol{y}_N$ in Equation (9). Formally, a state in the MDP is a tuple $s_t = (G, \mathcal{T}, S_t, t)$, where $G$ is a weighted graph of the problem, $\mathcal{T}$ the collection of set of terminals, and $S_t \subset V(G)$ is a partial solution constructed until the current timestep $t$ via previous actions. An action $a_t$ is to select a set of vertices $\boldsymbol{y}_t \subset V(G) \setminus S$ which includes a tree that spans the terminals $T_t \in \mathcal{T}$ as a subgraph of $G[\boldsymbol{y}_t]$. In turn, the subgraph $G[\boldsymbol{y}_t]$ is forwarded to the low-level agent which solves STP on the given subgraph by choosing the edges from $E(G[\boldsymbol{y}_t])$. Then, the high-level agent

---

[3]Since $|V| \approx O(\sqrt{|E|})$, so that $|Y| \ll |X|$ for large graphs.

receives negative of the sum of the edge weights of the low-level solution $L(\boldsymbol{y}_t)$ as a reward, and appends the solution $L(\boldsymbol{y}_t)$ to the previous partial solution $S_t$.[4] If the low-level solution $L(a_t)$ does not exist, or when any future choice of actions $a_{t+1}, \cdots, a_N$ leads to an infeasible solution of the given STPP, the high-level agent receives a large negative penalty $C < 0$. We defer detailed settings of our MDP formulation to the Appendix A.3.

**Model architecture**   Figure 2 shows the overview of our model. Since STPP is a CO problem defined over a weighted graph, we a use graph neural network (GNN) to encode the state representation with the policy network $\pi_\theta$ and the value function $V^{\pi_\theta}$ that serves as a baseline for actor-critic methods (Konda & Tsitsiklis, 1999).

Let $G$ be the weighted graph with edge weights $w_e$ as an edge feature for each $e \in E(G)$. First, a $D$-dimensional node feature $\boldsymbol{\mu}_v$ is computed for each node $v \in V$. Please refer to Appendix A.3 for our detailed choice of node and edge features. Then, the extracted features are encoded with graph attention network (GAT) (Veličković et al., 2018) and attention network (AT) (Vaswani et al., 2017). GAT aggregates the information across the *neighbors* in the graph to capture the local connectivity, but it is limited in modeling the long-range dependency Vaswani et al. (2017). We overcome the limitation by using the attention network. The attention network captures the long-range dependency by encoding relation between *all* (*i.e.*, ignores the graph structure) pairs of nodes. Global structures are further encoded via a graph embedding layer, which embeds particular subsets of node features into groups based on their characteristics. Detailed GNN architecture is provided in Appendix A.5. We use reinforcement learning to train high-level policy and the mathematical solver (*i.e.*, MILP solver) for solving the low-level problem (10) in our implementation[5].

## 4   Related Works

**Neural combinatorial optimization using reinforcement learning.**   Reinforcement learning (RL) approaches formulate the process of sequentially predicting the CO solution as a Markov decision process. To overcome the limitation of the supervised setting, Bello et al. (2017) proposed to train pointer network using policy gradient method by directly optimizing the cost without employing pre-computed solution. Subsequent works proposed to employ GNN (Khalil et al., 2017) and attention-based models (Kool et al., 2019; Nazari et al., 2018; Deudon et al., 2018) often combined with heuristic methods (Deudon et al., 2018) to solve other general CO problems with RL. However, existing works only focused on the CO instances where constraint can be easily satisfied by properly designing the action space (*e.g.*, TSP (Bello et al., 2017)). Our work is also RL-based approach, but overcome the limitation of RL approaches by hierarchically decomposing the problem. Intuitively, the decomposition of policy reduces the solution search space and facilitate the learning of feasible solution space.

**Combinatorial optimization with feasibility-hard constraints.**   There were few attempts to directly tackle CO problems with feasibility-hard constraints using RL. Ma et al. (2021) proposed to learn two separate RL models where the constraint satisfaction and objective optimization problems are respectively solved by each model. Cappart et al. (2021) manually shaped the reward to bias the RL process toward predicting feasible solution, and combined with constraint programming methods to guarantee the feasibility of solution. Our work indirectly tackles the feasibility-hard constraint by decomposing the given constraint satisfaction problem into two easier sub-problems with smaller problem size and search space so that the learning algorithm can efficiently solve each sub-problem.

**Decomposition of combinatorial optimization problem.**   Several decomposition methodologies have been introduced to address variety of CO challenges. Nowak-Vila et al. (2018) introduced a Divide-and-Conquer (DnC) framework focusing on scale-invariant problems, which assumes the problem can be split into sub-problems, solved independently, and subsequently merged to form a full solution. Hou et al. (2022) and Fu et al. (2021) proposed the domain specific approaches to divide problems and merging partial solutions

---

[4]Here, the low-level sub-problem can again be defined by a MDP $\mathcal{M}_{\text{lo}}$, which is equivalent to the original decision making process (3) but on a smaller problem instance $\text{span}(\phi^{-1}(a_t))$.

[5]Since our decomposition keeps the size of low-level subproblem small, we can run MILP within short time.

(via heatmap and MCTS) for VRP and TSP, respectively. Despite the utility, the applicability of such DnC-based methods is limited by their reliance on scale invariance, often leading to infeasible solutions when this assumption is violated. In a different vein, local search-based methods from Song et al. (2020) and Li et al. (2021) require an initial solution to iteratively refine decision variables, which is a notable constraint. Wang et al. (2021) further investigated a bi-level formulation, here the high-level problem is modifying the given problem instance and low-level problem is solving the modified instance, to ease problem solving. Yet, this occasionally resulted in more complex instances (*e.g.*, feasible solution may not exist) and suboptimal solutions. Our HCO also adopts the bi-level formulation but sidesteps these issues by preserving the original problem structure.

## 5 Experiments

We evaluate our model on STPP domain, one of the feasibility-hard CO problems. Firstly, we show our hierarchical model enhance the learning efficiency (Table 1 and Figure 3). Secondly, we verify our model can improve generalization performance on unseen larger instances. (Figure 4) Lastly, we examine the impact of the size of the pre-constructed dataset on learning through experimentation. (Figure 5) The rest of this section is structured as follows. First, we describe our experimental setup in Section 5.1 including the dataset construction, baselines, training, and evaluation. Then we present the experiment results in Section 5.2.

### 5.1 Setting

**Dataset**  To evaluate the learning efficiency and generalization capacity of each algorithm, we created a set of *feasibility-hard* STPP instances across multiple scales. Our instance generation protocol involved graphs of 40, 60, 80, and 100 nodes. For every graph category, we generated 50,000 instances for training, 1,000 for testing, and 100 for validation purposes. The graph structure was derived using the Watts Strogatz (WS) model (Watts & Strogatz, 1998) with the mean node degree $k \sim \text{Uniform}(\{3, 4, 5, 6\})$ and a rewiring probability $\beta \sim \text{Uniform}(0, 1)$. Weights are assigned to each edge from $[0, 1]$ uniformly at random. Given the random graph, a subset of vertices are chosen as the terminals (see the definition in Section 2.2) to construct the STPP instance. However, we found that randomly choosing the terminals mostly yields the instance that is either not solvable (*i.e.*, feasible solution does not exist) or trivially solvable (*i.e.*, removing constraint does not change the optimal solution). Thus, we designed the terminal node selection algorithm to ensure that the generated instances are solvable and non-trivial. We first partitioned the generated random graph graph into $N_{\text{type}}$ subgraphs using Lukes algorithm (Lukes, 1974), and then chose $N_{\text{terminal}}$ terminals within each partitioned subgraph. In particular, each subgraph is partitioned in a way such that a single tree that spans all the nodes in a subgraph exists. However, depending on the edge connectivity of the subgraph and the choice of terminals among the nodes, the resulting net may be trivial (*e.g.*, a net may consist of a small fraction of the partitioned subgraph). Thus, we carefully choose $N_{\text{type}}$ and $N_{\text{terminal}}$, and filter out the instances that are either unsolvable or trivial. For more detail, please refer to Appendix A.7.

**Baselines**  We compare our model with below baselines:

- MILP-$t$ uses the mixed integer linear programming (MILP) solver to find the best solution within given time limit $t$. We used OR-Tools (Perron & Furnon) in implementation.

- MILP-$\infty$ is the MILP solver without time limit, which finds the optimal solution.

- PathFinder (McMurchie & Ebeling, 1995) is a heuristic algorithm solving STPP (see Section 4 for more details). We used the publicly available implementation (Lee et al., 2022) with two variations of low-level solver: shortest-path finding (PathFinder-SP) and two-approximation (PathFinder-TA).

- Flat is a conventional (*i.e.*, non-hierarchical) RL agent that tries to maximize reward in the given MDP (*i.e.*, Section 2.1)

- Two-stage Divide Method (TAM) (Hou et al., 2022) decomposes the STPP problem in a one-shot manner; i.e., the problem is entirely decomposed first, and then each sub-problem is tackled

individually. This contrasts with HCO, which adopts an iterative approach to select and solve one sub-problem at a time. See Appendix A.4 for a detailed comparison of HCO with TAM.

For a fair comparison, we set $t=1$ second for MILP-$t$ to roughly match the execution time of the compared methods.

**Training** For HCO, we first pre-train them using behavioral cloning (Bain & Sammut, 1995) and then use IMPALA (Espeholt et al., 2018) for finetuning. We generate our behavioral cloning data using a solver (OR-Tools) and use cross entropy loss for training it. And for learning framework of RL, IMPALA of RLlib (Liang et al., 2017) is used. The hyper-parameters are chosen based on the performance on validation set. The chosen hyper-parameters and training method details are described in Appendices A.6 and A.8.

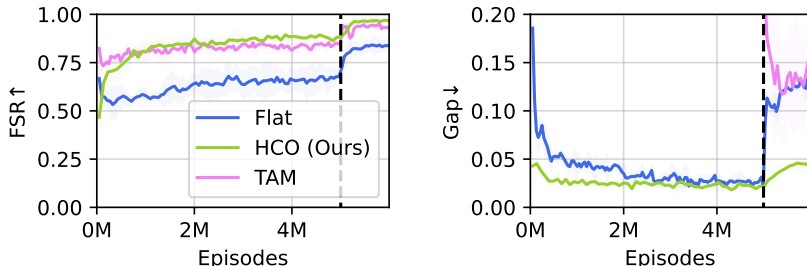

Figure 3: Training performance of the HCO for problem size $n = 40$. We pre-trained the agents via behavioral cloning until 5 millions episodes (*i.e.*, vertical dotted line in the figure), and then finetuned via reinforcement learning afterwards.

**Evaluation** We use three metrics to evaluate the algorithm's capability to minimize the cost and satisfy the constraint. *Feasible solution ratio (FSR)* is the ratio of instances where a feasible (*i.e.*, constraint is satisfied) solution was found by the method. Since the solution cost can be computed only for a feasible solution, we also introduce the metric *optimality gap* (Gap) measuring average of the cost suboptimality in feasible solutions found: $\text{Gap}=\left(\frac{\text{algorithm cost}}{\text{optimal cost}} - 1\right)$. Finally, *elapsed time (ET)* measures the average wall clock time taken to solve each instance in the test set. For reinforcement learning, we report the performance averaged over four random seeds.

## 5.2 Result

**Training Performance** Figure 3 shows the learning curves of HCO. Overall, HCO, Flat and TAM show similar results in terms of FSR but HCO (Ours) learn in sample efficient manner in Gap, due to smaller search space resulting from hierarchical decomposition. The agent is first trained using imitation learning until 500K episodes. The performance improves in terms of both FSR and Gap. Then, agent is updated using reinforcement learning method which directly minimizes the cost while trying to satisfy the constraint. The result shows that RL improves the performance of HCO, Flat and TAM. We note that performing RL from scratch makes the training significantly unstable since randomly initialized policy almost always generate an infeasible solution (*i.e.*, sparse reward problem).

**Generalization to Unseen Instances** Table 1 summarizes the performance of each method on unseen instances with *same* graph size. The first row represents HCO. We claim that our hierarchical framework improves the overall sample efficiency of reinforcement learning due to the reduced search space in high-level and low-level problems. MILP-1s achieves the near-optimal performance in terms of Gap for all the instance sizes, but the FSR quickly degrades as the instance size grows. This indicates that satisfying the constraint is much more challenging (*i.e.*, MILP-1s spends much more time) for MILP-1s than minimizing the cost. HCO outperforms Flat and TAM in terms of FSR and Gap thanks to the reduced search space. Regarding execution time (ET), TAM is more computationally efficient than HCO, owing to its one-shot problem partitioning approach, in contrast to HCO's iterative sub-problem selection that accounts for partial solution

Table 1: Result table for Steiner Tree Packing problem. Gap, and FSR mean the average optimality gap, and feasible solution ratio, respectively, and ET represents the average time taken to process a test instance.

| | $n = 40$ | | | | $n = 60$ | | |
|---|---|---|---|---|---|---|---|
| | Gap↓ | FSR↑ | ET (ms)↓ | | Gap↓ | FSR↑ | ET (ms)↓ |
| HCO (Ours) | **0.039** | **0.969** | 38 | HCO (Ours) | **0.031** | **0.953** | 99 |
| TAM | 0.105 | 0.885 | **17** | TAM | 0.128 | 0.914 | **46** |
| Flat | 0.045 | 0.957 | 106 | Flat | 0.087 | 0.935 | 252 |
| MILP-1s | 0.000 | 1.000 | 127 | MILP-1s | 0.000 | 0.982 | 501 |
| PathFinder-SP | 0.112 | 0.974 | 5 | PathFinder-SP | 0.165 | 0.990 | 8 |
| PathFinder-TA | 0.116 | 0.966 | 19 | PathFinder-TA | 0.147 | 0.974 | 48 |
| MILP-$\infty$ | 0.000 | 1.000 | 125 | MILP-$\infty$ | 0.000 | 1.000 | 532 |
| | $n = 80$ | | | | $n = 100$ | | |
| | Gap↓ | FSR↑ | ET (ms)↓ | | Gap↓ | FSR↑ | ET (ms)↓ |
| HCO (Ours) | **0.054** | **0.932** | 124 | HCO (Ours) | **0.056** | 0.892 | 246 |
| TAM | 0.246 | 0.620 | **47** | TAM | 0.291 | 0.520 | **47** |
| Flat | 0.087 | 0.902 | 529 | Flat | 0.062 | **0.905** | 679 |
| MILP-1s | 0.001 | 0.832 | 975 | MILP-1s | 0.000 | 0.035 | 1007 |
| PathFinder-SP | 0.150 | 0.976 | 20 | PathFinder-SP | 0.155 | 0.970 | 35 |
| PathFinder-TA | 0.149 | 0.965 | 115 | PathFinder-TA | 0.150 | 0.954 | 170 |
| MILP-$\infty$ | 0.000 | 1.000 | 1648 | MILP-$\infty$ | 0.000 | 1.000 | 4685 |

feasibility. We note that it ultimately contributes to improved FSR and Gap of HCO. Both PathFinder methods show overall slightly worse Gap but the highest FSR compared to other methods. We ascribe their high FSR to its iterative algorithm, *negotiated-congestion avoidance*, that is tailored for finding the feasible solution in STPP. Lastly, MILP-$\infty$ can find the optimal solution using tree search but the computation (*i.e.*, ET) increases exponentially in terms of the problem size ($n$), which is not scalable.

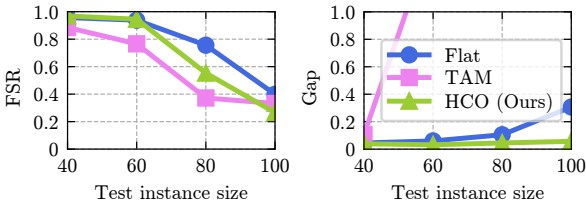
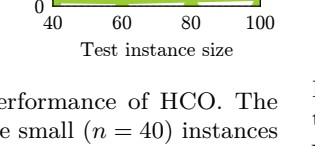
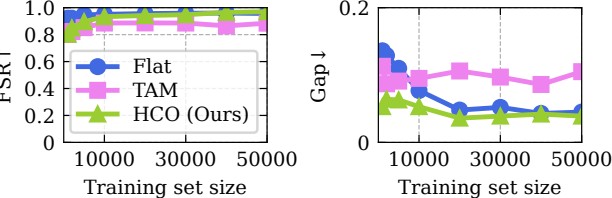

Figure 4: Generalization performance of HCO. The agent was trained only on the small ($n = 40$) instances and evaluated on the unseen and larger ($n \geq 40$) instances.

Figure 5: Analysis on the effect of training data set size on the performance. We trained all learning-based models on varying size of training set (x-axis) and evaluated on the test set in term of FSR (left) and Gap (right)
.

**Generalization to Unseen and Larger Instances** Figure 4 stress-tests how well the HCO, Flat and TAM can extrapolate to unseen and larger instances. Specifically, we trained the all learning-based models on a small ($n = 40$) instances and evaluated on a larger instances with the graph size $n \in \{40, 60, 80, 100\}$. As the graph size increases (*i.e.*, larger distribution shift in data), the performance generally decreases, but HCO maintains a reasonably low Gap, while Flat and TAM can't. We attribute it to our hierarchical decomposition framework that keeps the size of the sub-problem presented to the low-level agent consistent even when the instance size changes. Then, low-level agent will be affected less by the distribution shift and can generalize better.

**Effect of Training Set Size** We analyze the effect of training dataset size on the performance. We randomly sub-sampled $D$ instances from the 50,000 training instances of the size $n = 40$, trained the all methods on the sub-sampled data set, and evaluated on the entire test set of the size $n = 40$. We report the performance after training the agent with behavioral cloning on each sub-sampled training set for 100 epochs and RL until convergence. Figure 5 summarizes the result. Small datasets (less than 10,000 data) have poor FSR, but using more than 20,000 improves FSR and Gap score to match results from using 50,000 data for all methods. We also observe that the performance improves when the model is trained on larger number of instances. Unlike Flat and TAM, the HCO achieves near-optimal performance in term of Gap even with only 1,000 training instances. We note that Gap is measured *only* over the instances that the algorithm predicted a *feasible* solution. This indicates that HCO can easily minimize the cost with the help of MILP solver used in low-level agent if the high-level agent resolves the constraint properly (*i.e.*, find the feasible solution).

## 6 Conclusions

In this work, we proposed a novel hierarchical approach to tackle challenging CO problems with complex constraint. The central idea of the approach is to decompose the solution search space using latent mapping, resulting in a more sample-efficient learning due to separation of concerns and a smaller search space, as well as stronger generalization capability due to homogeneous problem size for low-level policy. To this end, a general hierarchical decomposition framework is formulated, which can be applied to any CO problem. The practical implementation of this framework is demonstrated for the specific case of the STPP using a hierarchical policy architecture and a graph neural network. The effectiveness of the proposed method is evaluated on large-scale STPP instances, and it is shown that the hierarchical framework improves the sample efficiency and generalization capability of the model, outperforming heuristic, mathematical optimization and learning-based algorithms specifically designed for STPP.

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

# A Appendix

## A.1 Combinatorial Optimization as Markov Decision Process

In this section, we provide a full illustration of a sequential optimization process for a general CO problem. Our goal is to design a corresponding MDP for Equation (3). Intuitively, the sequential decision making process in Equation (3) can be thought of choosing for each timestep $t$ an action $\boldsymbol{x}_t \in X_t$, until we have a full solution $\boldsymbol{x} = \sum_t \boldsymbol{x}_t$. However, note that we are constructing $X_1, \cdots, X_H$ *sequentially, i.e.*, we do not have the subspace decomposition $X = X_1 \oplus \cdots \oplus X_H$ beforehand. Hence, assume we have constructed $X_1, \cdots, X_t$ until the current timestep $t$. Let us define $W_t$ the remaining subspace that are yet to be decomposed, *i.e.*, $X = X_1 \oplus \cdots \oplus X_t \oplus W_t$. Then, the sequential decision making is equivalent to choosing for each timestep $t$ a subspace $X_t \leq W_t$ and consequently an action $\boldsymbol{x}_t \in X_t$, until we have a trivial subspace $W_t = \{\boldsymbol{0}\}$. We illustrate this process formally in Algorithm 1.

---

**Algorithm 1:** Sequential Optimization for Combinatorial Optimization in MDP

**Input:** Problem instance (1), stationary policy $\pi_{\text{lo}}$.
**Result:** Solution $\boldsymbol{x} = \sum_{t=1}^{T} \boldsymbol{x}_t$.

1   *initialize* $W_0 \leftarrow X$
2   *initialize* $A_0 \leftarrow \emptyset$           // Set of past actions.
3   **for** $t = 1, \cdots, H$ **do**
4      Update state $s_t \leftarrow (W_{t-1}, A_{t-1})$
5      Determine the action set $X_t \leq W_{t-1}$ such that $\dim(X_t) = 1$      // By policy or environment.
6      $\boldsymbol{x}_t \sim \pi_{\text{lo}}(s_t)$ and $A_t \leftarrow A_{t-1} \cup \{\boldsymbol{x}_t\}$      // Sampling $\boldsymbol{x}_t$ from $X_t$.
7      $W_t \leftarrow W$ such that $X = X_1 \oplus \cdots \oplus X_t \oplus W$
8   **end**

---

Sequential decision making process for the bi-level decomposition in Equation (7) and Equation (8) can be formulated similarly. The key is to construct the subspace decomposition $Y = Y_1 \oplus \cdots \oplus Y_N$ sequentially as done in Algorithm 1, while obtaining the partial solution for the original problem from the low-level sub-problem (8) simultaneously. Full illustration of the bi-level optimization is provided in Algorithm 2.

---

**Algorithm 2:** Sequential Bi-level Optimization for Combinatorial Optimization in MDP

**Input:** Problem instance (1), stationary high-level policy $\pi_{\text{hi}}$ and low-level policy $\pi_{\text{lo}}$.
**Result:** Solution $\boldsymbol{x} = \sum_{t=1}^{T} \boldsymbol{x}_t$.

1   *initialize* $W_0 \leftarrow X$ and $V_0 \leftarrow Y$
2   *initialize* $A_0 \leftarrow \emptyset$           // Set of past actions.
3   **for** $t = 1, \cdots, H$ **do**
4      Update state $s_t \leftarrow (V_{t-1}, A_{t-1})$
5      Determine the action set $Y_t \leq V_{t-1}$ such that $\dim(Y_t) = 1$      // By policy or environment.
6      $\mathcal{F}_t \leftarrow \left( \mathcal{F} \cap \left( \sum_{j=1}^{t-1} \boldsymbol{x}_j + W_{t-1} \right) \right) \Big|_{\text{span}(\phi^{-1}(Y_t))}$      // Local restriction of the feasible solution space.
7      $\boldsymbol{y}_t \sim \pi_{\text{hi}}(s_t)$ and $A_t \leftarrow A_{t-1} \cup \{\boldsymbol{y}_t\}$      // High-level problem
8      $X_t \leftarrow \text{span}(\phi^{-1}(\boldsymbol{y}_t)) \cap W_{t-1}$      // Search space for the low-level subproblem
9      $\boldsymbol{x}_t \leftarrow \text{L}(\boldsymbol{y}_t) := \arg\min_{\boldsymbol{x} \in X_t} \{ f(\boldsymbol{x}) : \boldsymbol{x} \in \mathcal{F}_t \}$      // Low-level solution from $\pi_{\text{lo}}$ and Algorithm 1.
10     $W_t \leftarrow W$ such that $X = \text{span}(\phi^{-1}(Y_1)) \oplus \cdots \oplus \text{span}(\phi^{-1}(Y_t)) \oplus W$
11     $V_t \leftarrow V \cap \phi(W_t)$ such that $Y = Y_1 \oplus \cdots \oplus Y_t \oplus V$
12   **end**

---

## A.2 Proof of Theorem 3.1

We begin with a proposition which we adopt as an important assumption of our latent mapping $\phi$.

**Proposition A.1.** *Let $X$ and $Y$ be $n$ and $m$-dimensional vector spaces over a finite field $F$, respectively. Then there exists a surjective latent mapping $\phi : X \to Y$ such that $|\mathrm{span}(\phi^{-1}(\boldsymbol{y}))| \leq O(|X|/|Y|)$ for all $\boldsymbol{y} \in Y$.*

*Proof.* In fact, the above property always holds if the mapping $\phi$ is linear. We claim that for a linear surjection $\phi : X \to Y$, $\dim(\mathrm{span}(\phi^{-1}(\boldsymbol{y}))) \leq n - m + 1$, and hence $|\mathrm{span}(\phi^{-1}(\boldsymbol{y}))| \leq |F|^{n-m+1} = O(|X|/|Y|)$. Let $B = \{\boldsymbol{v}_1, \cdots, \boldsymbol{v}_m\}$ be a basis for $Y$. Since $\phi$ is a surjection, there exists $\boldsymbol{u}_j \in X$ such that $\phi(\boldsymbol{u}_j) = \boldsymbol{v}_j$ for each $j = 1, \cdots, m$. Let $\boldsymbol{y} \in Y$ be arbitrary. Then there exist scalars $c_1, \cdots, c_m \in F$ such that $y = \sum_{j=1}^m c_j \boldsymbol{v}_j$. Then we have

$$\phi^{-1}(\boldsymbol{y}) = \{\boldsymbol{x} \in X : \phi(\boldsymbol{x}) = \boldsymbol{y}\} = \{\boldsymbol{x} \in X : \phi(\boldsymbol{x}) = \sum_{j=1}^m c_j \boldsymbol{v}_j\} \tag{11}$$

$$= \{\boldsymbol{x} \in X : \phi(\boldsymbol{x} - \sum_{j=1}^m c_j \boldsymbol{u}_j) = \boldsymbol{0}\} \subseteq \ker \phi + \sum_{j=1}^m c_j \boldsymbol{u}_j \tag{12}$$

where the third equality follows from the linearity of $\phi$. Note that since $\phi$ is a surjection, $\dim(\ker \phi) = n - m$ from the rank-nullity theorem. Thus, we have $\dim(\mathrm{span}(\phi^{-1}(\boldsymbol{y}))) \leq \dim(\mathrm{span}(\ker \phi + \sum_{j=1}^m c_j \boldsymbol{u}_j)) \leq n - m + 1$ as desired. $\qquad\square$

The existence of such $\phi$ is important, since we wish to adopt such a $\phi$ as our latent mapping to prove its usefulness. Note that if we are given arbitrary $\phi$, then our decomposed problem is as hard as the original problem. For instance, consider a mapping $\phi : X \to Y$ such that $|\phi^{-1}(y)| = 1$ for all $y \neq 0$ and $|\phi^{-1}(0)| = |X| - |Y| + 1$. In this case, the solution space of $\mathcal{M}_{\mathrm{lo}}$ is either trivial or as large as the original problem. Hence, for the remaining part of this section, we assume that our latent mapping $\phi$ is *well-chosen* as in Proposition A.1.

Before we prove our main theorem, we demonstrate how we can train our agents in $\mathcal{M}_{\mathrm{hi}}$ and $\mathcal{M}_{\mathrm{lo}}$ of our bi-level decomposition framework (Algorithm 3). Notice that we are exploiting the fact applying the optimal solver $L$ to a smaller subspace $\mathrm{span}(\phi^{-1}(\boldsymbol{y})) \leq X$ instead of entire $X$ is plausible for many problem settings. We first begin with a definition of a *block MDP*.

**Definition A.2** (Block MDP). A block MDP is defined by the tuple $(S, A, P, R, b, B, \gamma)$, where $S$ is a set of states, $A$ is a set of actions, $P : S \times A \to \Delta(S)$ is a transition function, $R : S \times A \to [0, 1]$ is a reward function, $B$ is a set of blocks, induced by a surjective blocking function $b : S \to B$, and $\gamma \in (0, 1]$ is a discount factor.

Finally, an important lemma from Kakade (2003):

**Lemma A.3** (Sample complexity lower bound; Kakade (2003)). *Assume that a $H$-step Markov decision process with state space $S$ and action space $A$ is given. Fix $\epsilon, \delta > 0$ and a state $s \in S$. Let $\mathcal{A}$ be any algorithm that has access only to a generative model of the MDP, and outputs a policy $\pi$ that satisfies $V_\pi(s) \geq V^*(s) - \epsilon$, with probability greater than $1 - \delta$. Then, the algorithm $\mathcal{A}$ must make at least $\Omega(\epsilon^{-1}|S||A|H \log(1/\delta))$ calls to the generative model of the MDP.*

*Proof.* See Theorem 2.5.2 and Theorem 8.3.4 of Kakade (2003). $\qquad\square$

We now provide a full proof of Theorem 3.1.

**Theorem 3.1** (Sample Complexity Reduction). *Let $\mathcal{M}$ be a $H$-step MDP as in (3). For any algorithm $\mathcal{A}$ that has access to $\mathcal{M}$ which outputs a policy $\pi_\theta$ such that $V^{\pi_\theta}(s) \geq V^*(s) - \epsilon$ with probability greater than $1 - \delta$ for a given state $s$, $\mathcal{A}$ must make at least $\Omega(\epsilon^{-1}|X|^2 H \log(1/\delta))$ calls of $\mathcal{M}$, whereas $\Omega(\epsilon^{-1} \max(|Y|, |X|/|Y|)^2 H \log(1/\delta))$ calls of either $\mathcal{M}_{hi}$ or $\mathcal{M}_{lo}$ is sufficient when using a bi-level decomposition described as in (7) and (8).*

---

**Algorithm 3:** Training high-level policy $\pi_{\mathrm{hi}}$ and low-level policy $\pi_{\mathrm{lo}}$ in Bi-level Framework

---

**Input:** Problem instance (1), and some constants $\epsilon, \delta > 0$.
**Result:** Near-optimal policies $\pi_{\mathrm{hi}}^*$ and $\pi_{\mathrm{lo}}^*$.

**1** Initialize randomized policies $\pi_{\mathrm{hi}}$ and $\pi_{\mathrm{lo}}$.
**2** Equip $\mathcal{M}_{\mathrm{hi}}$ defined as in Algorithm 2 with an optimal low-level solver $L(\cdot)$.
**3** Train $\pi_{\mathrm{hi}}$ on the above $\mathcal{M}_{\mathrm{hi}}$ to obtain deterministic near-optimal policy $\pi_{\mathrm{hi}}^*$. // Policy or value iteration, etc.
**4** Fix obtained high-level policy $\pi_{\mathrm{hi}}^*$.
**5** Replace optimal solver $L(\cdot)$ with $\pi_{\mathrm{lo}}$ in $\mathcal{M}_{\mathrm{hi}}$.
**6 for** $t = 1, \cdots, T$ **do**
**7** $\quad$ Call $\mathcal{M}_{\mathrm{hi}}$ to sample $\boldsymbol{y}_t \sim \pi_{\mathrm{hi}}^*(s_t)$, $W_{t-1}$, and $\mathcal{F}_t$ in Algorithm 2.
**8** $\quad$ Obtain $\mathcal{M}_{\mathrm{lo}}^{(t)}$ from $\mathcal{F}_t$ and $X_t = \mathrm{span}(\phi^{-1}(\boldsymbol{y}_t)) \cap W_{t-1}$.
**9 end**
**10** Train $\pi_{\mathrm{lo}}$ on a block MDP defined by collecting all $\mathcal{M}_{\mathrm{lo}}^{(t)}$ to obtain $\pi_{\mathrm{lo}}^*$.

---

*Proof.* Let $X$ and $Y$ be $n$ and $m$-dimensional vector space over a finite field $F$, respectively. Let $\mathcal{M}$ be a Markov decision process for Equation (3) as described in Algorithm 1. Since the state space consists of all possible partial solutions for $\boldsymbol{x}$, we have at most $2^n|X|$ states. The action $\boldsymbol{x}_t \in X_t$ in Algorithm 1 for any timestep $t$ is precisely a choice $\boldsymbol{x}_t \in F$. Thus, for any given $X$ over a finite field $F$, we have $|S||A| = 2^n|F|^{n+1} = O(|X|^2)$. This along with Lemma A.3 proves the first part of the theorem (*i.e.*, the lower bound for sample complexity of $\mathcal{M}$).

To prove the remaining part of Theorem 3.1, let $\mathcal{M}_{\mathrm{hi}}$ and $\mathcal{M}_{\mathrm{lo}}$ be MDPs for high-level problem (7) and corresponding low-level subproblem (8), respectively, where we train both $\mathcal{M}_{\mathrm{hi}}$ and $\mathcal{M}_{\mathrm{lo}}$ with Algorithm 3. The state space of $\mathcal{M}_{\mathrm{hi}}$ consists of all possible partial solutions in $Y$, and hence there are at most $2^m|Y|$ states. The action space is again isomorphic to the field $F$, so that we have $|S||A| = O(|Y|^2)$ as in the above claim. Finally, suppose we obtained $\pi_{\mathrm{hi}}^*$ from Algorithm 3. Note that low-level MDP $\mathcal{M}_{\mathrm{lo}}$ is equivalent to a block MDP in Algorithm 1 with its sub-MDPs collected from $\mathrm{span}(\phi^{-1}(\pi_{\mathrm{hi}}^*(s_t)))$ for each $t = 1, \cdots, H$. Since their sizes are bounded above with $O(|X|/|Y|)$ by Proposition A.1, any algorithm is expected to learn on $\mathcal{M}_{\mathrm{lo}}$ with sample complexity $\Omega(\epsilon^{-1}(|X|/|Y|)^2 H \log(1/\delta))$. As we are training $\pi_{\mathrm{hi}}$ and $\pi_{\mathrm{lo}}$ independently on $\mathcal{M}_{\mathrm{hi}}$ and $\mathcal{M}_{\mathrm{lo}}$, respectively, the overall sample complexity for the bi-level decomposition framework is additive. This completes the proof. $\qquad\square$

### A.3 MDP formulation

Below we provide a full description of our MDP formulation for STPP in Section 3.3.

**State** The state $s_t$ of the high-level MDP $\mathcal{M}_{\mathrm{hi}}$ consists of a graph $G$, collection of set of terminals $\mathcal{T}$, a partial solution $S_t \subset V(G)$ constructed until timestep $t$, (*i.e.*, the nodes of the disjoint trees that span terminals $T_1, \cdots, T_{t-1}$) and the current timestep $t$. The graph $G$ provides a general information of the problem instance, *i.e.*, the connectivity of the graph via edges. In practice, the graph $G$ can be represented by the adjacency matrix $A \in \mathbb{R}^{|V| \times |V|}$, where $a_{ij}$ takes the edge weight $w_{ij}$. The information can be further encoded via message passing layers of GAT and AT in GNN. $\mathcal{T}$, $S_t$ and $t$ provide node features for current timestep, and are essential for generating a graph embedding. From the state information $s_t$, we extract the node features of the graph to encode further via GNN model. For a node $v \in V$, we denote the node features of the vertice $v$ as $\mathbf{x}_v := (\mathrm{x}_o, \mathrm{x}_\tau, \mathrm{x}_d) \in \mathbb{Z}^3$. The first node feature $\mathrm{x}_o \in \{0, 1\}$ denotes whether a node $v$ is included in the current partial solution or not. If $v$ is selected as a partial solution, we define $\mathrm{x}_o = 1$, and otherwise 0. The second node feature $\mathrm{x}_\tau \in \{0, 1, \cdots, N_{\mathrm{type}}\}$ indicates the terminal type (*i.e.*, $\mathrm{x}_\tau = k$ if and only if $v \in T_k$), where the indices are labeled in the order that the high-level MDP solves for. Non-terminal nodes will be assigned a value of 0. The last feature, $\mathrm{x}_d$ denotes the degree of a node $v \in V$. The edges of the

Table 2: Node features

| Notation | Value | Description |
|---|---|---|
| $\mathrm{x}_o$ | $\in \{0, 1\}$ | Is partial solution |
| $\mathrm{x}_\tau$ | $\in \{0, 1, 2, ..., N_\text{type}\}$ | Terminal type |
| $\mathrm{x}_d$ | $\in \mathbb{Z}$ | Node degree |

Table 3: Edge feature

| Notation | Value | Description |
|---|---|---|
| $w_e$ | $\in \mathbb{R}$ | Edge weight(=cost) |

graph are also assigned with edge features. We only use the edge weights as the edge features in this paper. Our choice of node and edge features are summarized in Tables 2 and 3.

**Action** The action $a_t$ of a high-level MDP $\mathcal{M}_\text{hi}$ at timestep $t$ is to select a set of vertices $\boldsymbol{y}_t \subset V(G) \setminus S_t$, where $S_t$ is the partial solution constructed until the previous timesteps. Intuitively, the action $a_t$ is to select a node *set* that includes all terminals of current type at timestep $t$. The selected set of nodes should create a subgraph $G[a_t]$ of $G$ generated by the nodes $a_t$, and should corrrespond to a STP instance (with single type of terminals). The low-level agent then computes a minimum weight tree that spans all terminals of current type from the subgraph $G[a_t]$. In practice, we further assist the agent by designing the MDP environment for $\mathcal{M}_\text{hi}$ in a way such that the terminal nodes of the current type are automatically selected by the environment internally. Hence, the action $a_t$ will result in a subgraph $G[a_t \cup V(T_t)]$ instead of $G[a_t]$.

**Reward** Designing a reward with optimality guarantee is a non-trivial task. In particular, we wish to construct a reward where all feasible solutions result in higher reward than those of any infeasible solutions. Also, the feasible solutions with *better* solution quality (*i.e.* objective being closer to optimal solution) should be assigned with higher reward. Hence, given a final solution $\boldsymbol{x}$ of a CO problem (1), a reward with optimality guarantee can be compactly formulated as $r(\boldsymbol{x}) = c \cdot \mathbf{1}_\mathcal{F}(\boldsymbol{x}) - f(\boldsymbol{x})$, where $c \geq \sup_{\boldsymbol{x} \in X} f(\boldsymbol{x})$, $f$ is the objective function of problem (1), and $\mathbf{1}_\mathcal{F}(\cdot)$ is an indicator function. Note that this form of reward is equivalent to what is described in Section 2.1; achieving the maximal return will result in the same optimal policy[6]. Finally, recall that our objective $f$ is linear of $X$; for each partial solution $\boldsymbol{x}_t$ constructed at timestep $t$, we are able to decompose our reward function as follows.

$$r(\boldsymbol{x}_t) = c \cdot \mathbf{1}_\mathcal{F} \left( \sum_{k=1}^{t} \boldsymbol{x}_k \right) - f(\boldsymbol{x}_t) \tag{13}$$

**Transition** Our transition in MDP is deterministic; the change in the partial solution alters a node feature $\mathrm{x}_o$ from 0 into 1, which result in different node and graph embeddings from GNN.

**Termination** Our STPP environment is terminated when it is not able to generate feasible STPP solution or when a feasible solution is found. The cases where generating a feasible STPP solution includes (1) no possible actions remaining, (2) any choice of actions in future timestep inevitably results in an infeasible solution, or (3) the choice of action $a_t$ in a high-level MDP $\mathcal{M}_\text{hi}$ results in an unsolvable STP instance $G[a_t]$.

### A.4 Comparison with TAM

Since there are no known neural methods that is designed to solve STPP, we establish minor modifications to TAM (Hou et al., 2022), one of the state-of-the-art neural method for solving the VRP problem, to be able to tackle STPP. Although TAM is algorithmically similar to HCO, the biggest difference is that the solutions

---

[6]Providing *incentives* for $\boldsymbol{x} \in \mathcal{F}$ instead of a penalty when $\boldsymbol{x} \notin \mathcal{F}$ scales all rewards to be non-negative.

to other sub-problems are not known. TAM consists of one high-level step followed by one low-level step, where problems divided in the first high-level step followed by one low-level step, where problems divided in the first high-level step are solved independently in the subsequent low-level step. In other words, while HCO learns to solve problems *sequentially*, TAM aims to first decompose the problem beforehand, and solve the subproblems simultaneously. For application in STPP, we structure it such that in the first step, all terminals are divided at once into subgraphs, and then each subgraph representing an STP problem is solved independently by MILP.

### A.5 GNN architecture

**Encoder** Given a graph $G$, we first extract a $D$-dimensional node embedding $\boldsymbol{\mu}_v$ for each node $v \in V$, where $D$ denotes the number of features provided as in Appendix A.3. Note that we use $D = 3$, which consists of $\mathbf{x}_v = (\mathrm{x}_o, \mathrm{x}_\tau, \mathrm{x}_d)$ respectively as in Table 2. Let $\rho : \mathbb{R}^D \to \mathbb{R}^{D \cdot p}$ be a fixed vectorization mapping of a given node feature $\mathbf{x}_v$, and let $\theta_0 : \mathbb{R}^{D \cdot p} \to \mathbb{R}^p$ be a linear mapping. Then we obtain the initial node embedding $\boldsymbol{\mu}_v$ as follows.

$$\boldsymbol{\mu}_v = \mathrm{ReLU}(\theta_0(\rho(\mathbf{x}_v))). \tag{14}$$

Then we further encode the node embeddings $\mathbf{M} := (\boldsymbol{\mu}_1, \cdots, \boldsymbol{\mu}_{|V|}) \in \mathbb{R}^{|V| \times p}$ via graph attention network (GAT) and attention network (AT). Formally, let $\Theta_i : \mathbb{R}^{n_h \times p} \to \mathbb{R}^{2p}$ and $\theta_i : \mathbb{R}^{2p} \to \mathbb{R}^p$ for $i = 1, \cdots, l$ be linear mappings, where $n_h$ denotes the number of heads of GAT. Let us slightly overload the notation and write $\Theta_i(\mathbf{M}) := (\Theta_i(\boldsymbol{\mu}_1), \cdots, \Theta_i(\boldsymbol{\mu}_{|V|}))$ and $\theta_i$ likewise. Then, we encode the node embedding recursively as follows.

$$\mathbf{M}^{(i-1)\prime} = \mathrm{AT}(\Theta_i(\mathrm{GAT}(\mathbf{M}^{(i-1)}; G)); G) \tag{15}$$

$$\mathbf{M}^{(i)} = \theta_i(\mathrm{ReLU}(\mathbf{M}^{(i-1)}||\mathbf{M}^{(i-1)\prime})) \tag{16}$$

for each $i = 1, \cdots, l$, where we define $\mathbf{M}^{(0)} \equiv \mathbf{M}$, and write $(\cdot||\cdot)$ for `CONCATENATE`$(\cdot, \cdot)$. In particular, we write our graph encoder function **Enc** briefly as follows, with some details omitted.

$$\mathbf{Enc}(\cdot; G) := \overbrace{(\mathrm{AT} \circ \mathrm{GAT}) \circ \cdots \circ (\mathrm{AT} \circ \mathrm{GAT})}^{l \text{ times}}(\cdot; G) \tag{17}$$

**Graph embedding, logit and probability** Given the node embedding of the last layer $\mathbf{M}^{(l)}$, we obtain the embedding for the entire graph $\boldsymbol{\mu}^G$ (see Figure 2). Instead of simply averaging over all the nodes, we enrich the graph embedding by grouping the nodes into three subsets based on their characteristics: current terminal $T_t$, partial solution $S_t$, and non-terminal nodes $\overline{V} := V \setminus \bigcup_{T \in \mathcal{T}} T$. Then, the embeddings are averaged within each subset, concatenated, and projected to obtain the graph embedding $\boldsymbol{\mu}^G$. Formally, the graph embedding layer **Emb** is written as follows.

$$\mathbf{Emb}(\cdot; t) := \Psi(\cdot, T_t)||\Psi(\cdot, S_t)||\Psi(\cdot, \overline{V}) \tag{18}$$

where $\Psi(\mathbf{M}^{(l)}, A)$ performs the average pooling over the set of node embeddings that belong to $A$. The graph embedding $\boldsymbol{\mu}^G$ is obtained as $\boldsymbol{\mu}^G = \mathbf{Emb}(\mathbf{M}^{(l)}; t)$. Finally, the logit value (*i.e.*, the probability of choosing the node) for each node $v \in V$ is computed as follows.

$$\mathrm{logit}_v = w_4(w_3(\mathrm{ReLU}(w_1(\boldsymbol{\mu}^G)||w_2(\boldsymbol{\mu}_v^{(l)}))) + \boldsymbol{\mu}_v^{(0)}) \quad \forall v \in V \tag{19}$$

$$p_v = \mathrm{softmax}(\mathrm{logit}_v) \quad \forall v \in V \tag{20}$$

where $w_1 : \mathbb{R}^{3p} \to \mathbb{R}^p$, $w_2 : \mathbb{R}^p \to \mathbb{R}^p$, $w_3 : \mathbb{R}^{2p} \to \mathbb{R}^p$, and $w_4 : \mathbb{R}^p \to \mathbb{R}^2$ are linear functions.

**Value function** The value function $V^{\pi_\theta}$ uses a model that has a similar GNN architecture with a simple multi layer perceptron MLP that sequentially projects ($\mathbb{R}^{3p} \to \mathbb{R}^{3p} \to \mathbb{R}^p \to \mathbb{R}^1$), and does not share weights from policy network.

$$V^{\pi_\theta}(s_t) = \mathrm{MLP}(\mathbf{Emb}(\mathbf{Enc}(\mathbf{M}; G); t)). \tag{21}$$

### A.6 Hyperparameters

The GNN model has a hidden dimension of $p = 128$, and $l = 5$ for GAT and AT encoder layers. Both GAT and AT use 8 heads, and the dropout rate is 0.5 in IL but not used in RL training. A batch size of 64 is used, and the learning rate is initialized to $10^{-4}$, which decreases by 0.99 per epoch. Fine-tuned value if $5 \times 10^{-7}$ is used for weight decay. To prevent divergence of learning, clips the gradient norm to 1. HCO learns 100 epochs, using 1 epoch as updating model with BC data in every step using every episode. In RL phase, batch size of 30, learning rate of $10^{-6}$, discount factor of 0.99, and an entropy coefficient of 0.01 are used, and the value function loss coefficient is set to 5. The number of workers used in IMPALA is set to 30.

### A.7 Dataset generation

Let $n$ be the number of nodes of a graph instance $G$. To assign terminal nodes while ensuring the existence of a feasible solution, we first set maximum number of type $M = 5$. And the number of terminal types $N_{\text{type}}$ is determined by (Lukes, 1974). The algorithm generates partitions of the graph with the determined $N_{\text{type}}$. The partitions are guaranteed to include a spanning tree by the algorithm, but their sizes may not be consistent, due to the randomness in Lukes algorithm. Finally, we randomly choose for each graph partition $N_{\text{terminal}} = \max(2, \lfloor n \times q/N_{\text{type}} \rfloor)$ terminal nodes from uniform distribution, to ensure that a net is formed (*i.e.*, a feasible solution exists), where $q = 0.2$ that follows convention from Yan et al. (2021). Through this way, it is possible to create a solution where a feasible solution exists, but to create feasibility-hard instances, instances that is solvable by sequential STP is excluded. Note that the number of terminal types $N_{\text{type}}$ is precisely the maximal length of the horizon of HCO MDP formulation.

### A.8 Training and evaluation

**Imitation learning for HCO agent.** For imitation learning, we first collect the demonstration data using the optimal solver, MILP, as expert policy. The expert policy $\pi^{\text{expert}}$ is defined as $\pi^{\text{expert}}(a_t|\cdot) = 1$ if $a_t \in \mathcal{A}_t^*$ and $\pi^{\text{expert}}(a_t|\cdot) = 0$ otherwise. The nodes that are not selected by the expert are excluded from the loss calculation. HCO use cross-entropy loss for BC training, and 1 epoch is defined as updating HCO for every step of every instance. During the evaluation phase, the softmax function is applied to the logit values of each node to obtain probabilities, and nodes with probability values exceeding 0.5 are selected as high-level actions. The training and evaluation are carried out on a single GPU, comprising an AMD EPYC 7R32 CPU and NVIDIA A10G GPU.

