# OpenReview forum: "Hierarchical Decomposition Framework for Feasibility-hard Combinatorial Optimization"
_TMLR — Rejected by TMLR_

### Review · Reviewer_MeyQ · 2023-11-27

**Summary Of Contributions:**

This paper proposes a hierarchical combinatorial optimizer (HCO) framework to tackle combinatorial optimization (CO) problems with complex feasibility-hard constraints. The key idea is to decompose the original CO problem into a high-level problem that suggests sub-problems, and a low-level problem that solves the suggested sub-problems. This decomposition results in smaller search spaces at both levels, facilitating more efficient learning. The authors demonstrate the framework on the Steiner tree packing problem and show that HCO outperforms several baselines including mathematical optimization and learning-based methods.

**Audience:**

Yes

**Claims And Evidence:**

No

**Requested Changes:**

- Why is Flat slower (i.e. has higher ET) than HCO?
- Ideally run experiments with multiple seeds to understand variance and check whether the performance differences are significant.
- Hopefully the code to reproduce the experiments could be released allowing the community to build on top of this setup.
- Would be good to explicitly state the limitations.

Minor:

- Figure 3 (right) is somewhat confusing. Might be useful to give a short takeaway message in the caption.
- Pg 3: "and does not learn efficiently" should be "and do not learn efficiently".

**Strengths And Weaknesses:**

**Strengths**

- The paper addresses an important class of challenge in CO: feasibility hard constraints where ensuring feasibility is not trivial.
- The hierarchical decomposition is fairly intuitive and the bilevel framing is well utilized to reduce search space/action space enabling more efficient learning
- Empirical results show better optimization performance as well reasonable generalization comoared to learning based baselines

**Weaknesses**

- Many details are somewhat unclear when it comes to broader takeaways. The decomposition seems somewhat problem specific and hard to say how broadly applicable the specific choices are to other CO problems with similar constraints.
- The NN design choices are completely opaque and lack any ablations to validate design choices. A lot of it seems unnecessary and I would expect something simpler to work equally well given the current choice of lower level policy.
- Baselines are somewhat less clear. Flat baseline seems to do quite well and given the lack of details, the performance difference could very well be due to suboptimal hyperparameters. In general, the results lack any error bars which given the use of RL make it very difficult to draw very strong conclusions.

---

> ### Author Response · Authors · 2024-01-29
> **Response**
>
> **1. Weaknesses**
>
> * Many details are somewhat unclear when it comes to broader takeaways. The decomposition seems somewhat problem specific and hard to say how broadly applicable the specific choices are to other CO problems with similar constraints.
>
> → Please see common response # 3 above.
>
>
> * The NN design choices are completely opaque and lack any ablations to validate design choices. A lot of it seems unnecessary and I would expect something simpler to work equally well given the current choice of lower level policy.
>
> → Please see common response # 1 above.
>
>
> * Baselines are somewhat less clear. Flat baseline seems to do quite well and given the lack of details, the performance difference could very well be due to suboptimal hyperparameters. In general, the results lack any error bars which given the use of RL make it very difficult to draw very strong conclusions.
>
> → We ensured uniformity in hyperparameters across Flat, HCO, and TAM, with the hyperparameters being set based on the Flat baseline. Specifically, in the experiments with 40 nodes, we considered the following hyperparameter ranges:
>
> p = {32, 64, 128, 256}
>
> l = {3, 4, 5}
>
> number of heads = {4, 8}
>
> IL learning rate = {1e-3, 1e-4, 1e-5, 1e-6}
>
> weight decay = {1e-6, 5e-7}
>
> RL learning rate = {1e-3, 1e-4, 1e-5, 1e-6}
>
> entropy coefficient = {1, 1e-1, 1e-2}
>
> value loss coefficient = {1e-1, 1, 3, 5}
>
> We began by randomly choosing 30 combinations of hyperparameters from the given range. Next, we identified the most effective combination by selecting the one that yielded the highest FSR performance using the **Flat** method on the validation set.
> Then, we used the same hyperparameter across all models. Details about the selected hyperparameters can be found in Appendix A.6. Empirically, we believe our performance is not merely a result of hyperparameter search. Please refer common response # 2 for multiple seed results. Considering the small standard deviation, the performance gap between HCO and Flat is significant.
>
>
> **2. Requested Changes**
>
> * Why is Flat slower (i.e. has higher ET) than HCO?
>
> → The reason why Flat is slower (i.e., has a higher Execution Time or ET) than HCO is due to the larger number of GNN forward passes involved in the Flat approach. Flat starts from a terminal node and selects neighborhood nodes **one by one**. On the other hand, HCO obtains the selection probabilities for all nodes of one type of terminal node in a single GNN forward pass and uses these probabilities to select **multiple nodes** at a time. Therefore, compared to HCO, Flat requires many more GNN forward passes to arrive at a solution, resulting in the slower speed in obtaining a solution.
>
>
> * Ideally run experiments with multiple seeds to understand variance and check whether the performance differences are significant.
>
> → We have reconducted experiments with multiple seed and the results are in common response # 2 above.
>
> * Hopefully the code to reproduce the experiments could be released allowing the community to build on top of this setup.
>
> → We plan to organize and share the code after the publication of the paper so that our codebase can be a good starting point for other feasibility-hard CO research.
>
> * Would be good to explicitly state the limitations.
>
> → In this research, we sought to define and illustrate feasibility-hard combinatorial optimization (CO) problems, which have not been extensively addressed in existing CO studies, using the Steiner Tree Packing Problem (STPP) as an example. However, we faced challenges in objectively demonstrating the extent of feasibility. For instance, in STPP, as the number or types of terminal nodes increase, the number of feasible solutions decreases, thereby elevating the importance of feasibility. Conversely, when there are fewer terminal nodes or types, the abundance of feasible solutions reduces the significance of feasibility. The complexity of feasibility can also vary with the same number of terminal nodes or types, depending on the graph's distribution, such as the number and connectivity of nodes and edges, posing difficulties in establishing an objective standard for feasibility difficulty. To address this, our study assumed that problems that remained unsolved by the low-level policy (i.e., STP) when tackled sequentially were of the same difficulty level. For future work, we anticipate that if a range of feasibility-hard CO problems can be defined and a difficulty scale between optimality and feasibility can be established, it will enable a selective approach to both aspects when solving real-world CO problems.

---

> ### Author Response · Authors · 2024-01-29
> **Response**
>
> **3. Minor**
>
> * Figure 3 (right) is somewhat confusing. Might be useful to give a short takeaway message in the caption.
>
> → Figure 3 (right) illustrates the learning curve of the compared methods in terms of optimality gap (Gap). We will add the following description about the detailed interpretation of the results in figure 3 to the caption.
>
> “Overall, the HCO achieves the smallest Gap with highest FSR, indicating efficient policy improvement for solving feasibility-hard CO problems.”
>
> Also we will add the detailed interpretation of Figure 3 (left) and (right) in Section 5.3 as follows:
>
> "The supervised pre-training improves both FSR and Gap of all the methods but the performance improvement plateaus around 5M steps. During RL finetuning, the FSR rapidly improves for all the methods while Gap worsens. During fine-tuning, a rise in FSR, indicating more feasible solutions, correlates with a growing gap. This suggests that instances solved later in training often need more training to reach optimality. Consequently, as FSR rises sharply, the model encounters many new instances for which it initially only finds suboptimal solutions, leading to an increase in Gap.”
>
> * Pg 3: "and does not learn efficiently" should be "and do not learn efficiently".
>
> → We will make the correction. Thank you.

---

### Review · Reviewer_yvFN · 2023-12-18

**Summary Of Contributions:**

This paper proposes an imitation+reinforcement learning approach along with hierarchical decomposition for efficiently solving combinatorial optimization problems. This is the first work to specifically solve the Steiner Tree Packing Problem (STPP) using an end-to-end learning framework.

**Audience:**

Yes

**Claims And Evidence:**

Yes

**Requested Changes:**

1. A broader applicability assessment, specifically, extending the study to include more types of CO problems to demonstrate the approach's versatility.
2. Having at least 6 random seeds for the result.
3. Correcting some citation format issues: e.g. Karp (1972) in page 3, Vaswani et al. (2017) in page 6.

**Strengths And Weaknesses:**

Strengths:
1. Innovative approach combining hierarchical decomposition and neural combinatorial optimization.
2. Empirical evidence of effectiveness over traditional heuristics and learning-based algorithms in tackling the STPP.
3. Enhanced learning efficiency and generalization capabilities in optimization tasks.

Weaknesses:
1. The focus is on a specific type of CO problem (STPP), potentially limiting broader applicability.
2. The assumption of hierarchical decomposition might introduce challenges in scalability or adaptability to different types of CO problems. Many CO problems cannot be decomposed similarly.

---

> ### Author Response · Authors · 2024-01-29
> **Response**
>
> **1. Weaknesses**
>
> * The focus is on a specific type of CO problem (STPP), potentially limiting broader applicability.
> * The assumption of hierarchical decomposition might introduce challenges in scalability or adaptability to different types of CO problems. Many CO problems cannot be decomposed similarly.
>
> → Please see common response # 3 above.
>
> **2. Requested Changes**
>
> * A broader applicability assessment, specifically, extending the study to include more types of CO problems to demonstrate the approach's versatility.
>
> → Implementing and experimenting with other feasibility-hard CO problems would currently require a significant investment of time and resources, which we may not be able to afford at the moment. But we provide application of possible bi-level formulation for other CO problems in common response # 3 above.
>
> * Having at least 6 random seeds for the result.
>
> → Please see common response # 2 above.
>
> * Correcting some citation format issues: e.g. Karp (1972) in page 3, Vaswani et al. (2017) in page 6.
>
> → We will make the correction. Thank you.

---

### Review · Reviewer_MbwE · 2024-01-16

**Summary Of Contributions:**

This paper provides a machine learning-based algorithm to solve combinatorial optimization problems that (a) decomposes the combinational optimization problem into a bi-level problem and (b) solves this bi-level formulation as a sequential problem using RL. The paper focuses specifically on the Steiner Tree Packing Problem (STPP). Experimental results on graphs of 40-100 nodes show that the proposed algorithm (HCO) improves solution feasibility and optimality gap compared to other methods, while running in a reasonable amount of time.

**Audience:**

Yes

**Claims And Evidence:**

No

**Requested Changes:**

Critical changes:
* Examine the impact of behavior cloning on the performance of the HCO method (e.g., via ablations and/or additional experiments), per the discussion above in "Weaknesses."
* Discuss in more depth why the proposed method is particularly well-suited to addressing the feasibility problem.
* Discuss or demonstrate the impact of the choice of model architecture - e.g. via discussion of whether baseline methods use comparably strong/expressive model architectures, or via ablation studies.
* Remove claims about improved generalization performance and data efficiency, or improve experimental results on these metrics.

Minor changes:
* Grammar and usage errors should be corrected in the "Experiments" section.

**Strengths And Weaknesses:**

Strengths:
* The explanation of the algorithm and its application to the Steiner Tree Packing Problem are exceedingly well done. I am not an expert in combinatorial optimization but still found the explanation very clear.
* The algorithm is clever and well-suited to the structure of the Steiner Tree Packing Problem, significantly reducing complexity by searching over vertex space rather than edge space.
* The experimental setup is well-described and transparent.
* Results of the algorithm seem promising with respect to solution feasibility and optimality gap.

Weaknesses:
* The paper is motivated by the fact that prior work cannot properly deal with infeasible solutions. While the experiments show empirical improvement on this front, more intuition needs to be given regarding why the proposed method addresses this problem. Notably, other methods also seem to perform reasonably with respect to feasibility in the experimental results.
* The specific choice of model architecture is not justified, nor is it described whether baseline methods use comparably strong/expressive model architectures.
* The HCO method as demonstrated uses behavior cloning for 500K episodes. Behavior cloning requires solved instances (a limitation of supervised learning approaches, as stated in the intro). Much of the improvement of the HCO algorithm seems to be attributed to behavior cloning (in Figure 3, HCO has already achieved performance comparable to the other methods with behavior cloning), and statements about the data efficiency of HCO are dependent on what information was available/leaked during behavior cloning. Much more needs to be done to demonstrate the impact of behavior cloning, and separate the impact of this design choice from the impact of the proposed algorithm.
* Generalization performance of the algorithm is similar to, rather than improving on, baselines. (While Gap technically improves, FSR generalization performance across all methods is low enough that Gap is hard to interpret - it may refer to performance, e.g., on inherently easy problem instances.)
* Data efficiency performance of the algorithm is similar to, rather than improving on, baselines.

---

> ### Author Response · Authors · 2024-01-29
> **Response**
>
> **Weaknesses and requested Changes**
>
> * The paper is motivated by the fact that prior work cannot properly deal with infeasible solutions. While the experiments show empirical improvement on this front, more intuition needs to be given regarding why the proposed method addresses this problem. Notably, other methods also seem to perform reasonably with respect to feasibility in the experimental results.
> * Discuss in more depth why the proposed method is particularly well-suited to addressing the feasibility problem.
>
> → Our paper introduces the concept of feasibility-hard problems, which demand a dual focus on optimality and feasibility, inherently complicating the decision-making process in RL and impacting sample efficiency. To address this, we implement hierarchical reinforcement learning, strategically dividing the decision-making process into high-level and low-level policies. This division allows the high-level policy to concentrate on feasibility, while the low-level policy employs efficient heuristics to manage the optimality gap, significantly improving sample efficiency.
>
> Consider a deterministic low-level policy, such as the use of a heuristic algorithm in our experiment, or the greedy action selection (i.e., choosing the action with the highest probability) from a learned low-level policy. In these scenarios, the low-level solution space $\phi^{-1}(y)$, as determined by the high-level action, is effectively narrowed down to a specific low-level solution $L(y)$. Hence, the original condition $\phi^{-1}(y) \cap \mathcal{F} \neq \emptyset$ of the high-level problem reduces to $\phi^{-1}(y) \subset \mathcal{F}$ as follows:
>
> $$
> \begin{align}
> \underset{\boldsymbol{y} \in Y}{\mathrm{argmin}} \{\sum_{n \leq N}f(L(\boldsymbol{y})): \phi^{-1}(\boldsymbol{y}) \subset \mathcal{F} \}\\\\
> \text{where}, \ \mathrm{L}(\boldsymbol{y}) :=  \underset{\boldsymbol{x} \in \phi^{-1}(\boldsymbol{y})}{\mathrm{argmin}}f(\boldsymbol{x}).
> \end{align}
> $$
>
>
> Note that the low-level problem (i.e., second line of the above equation) now does not concern the feasibility of solution x, but only aims to optimize the cost. This clearly shows that our formulation allows the separation of concern between satisfying the feasibility constraint and optimizing the cost.
>
>
> * The specific choice of model architecture is not justified, nor is it described whether baseline methods use comparably strong/expressive model architectures.
> * Discuss or demonstrate the impact of the choice of model architecture - e.g. via discussion of whether baseline methods use comparably strong/expressive model architectures, or via ablation studies.
>
> → Please see common response # 1 above.

---

> ### Author Response · Authors · 2024-01-29
> **Response**
>
> * The impact of Behavior Cloning on the effectiveness and data efficiency of the HCO.
>
> → To evaluate the impact of behavior cloning in training HCO, we conducted additional experiment that trains HCO with RL from scratch, **without behavior cloning pretraining**. The model training is still ongoing, and we will further update with the complete result soon. From our early observation, we see the clear suboptimality in FSR performance, particularly noticeable in the initial stages of RL training. This challenge arises from the inherent difficulty of generating feasible solutions through random policies in feasibility-hard combinatorial optimization (CO) problems, which significantly impedes the learning process due to the absence of a learning signal—an issue not prevalent in less complex or non-feasibility-hard CO problems.
>
> On the other hand, Figure 3 in the paper reveals that relying solely on behavior cloning also fails to achieve optimal results, with FSR performance plateauing after approximately 4 million steps for all three compared methods (HCO, Flat, and TAM). This limitation underscores the need for RL to introduce additional learning signals and enhance performance.
>
> Therefore, integrating behavior cloning with subsequent RL finetuning emerges as a crucial strategy for attaining competitive results, combining the strengths of both approaches to overcome the inherent challenges of feasibility-hard CO problems.
>
>
> * Generalization performance of the algorithm is similar to, rather than improving on, baselines. (While Gap technically improves, FSR generalization performance across all methods is low enough that Gap is hard to interpret - it may refer to performance, e.g., on inherently easy problem instances.)
>
> → We acknowledge the difficulty in evaluating generalization performance across algorithms due to the intricate interplay between Gap and Feasible Solution Ratio (FSR). To facilitate a clearer comparison, we introduced the Common Instance Gap (CIG) metric, which evaluates the Gap exclusively on instances where every compared method identified a feasible solution. This approach ensures a fair comparison ground.
>
> Our refined generalization experiment, conducted with training on $n=40$ instances and testing across $n={40, 60, 80, 100}$, by measuring this CIG metric. We present the CIG performance, averaged across four distinct random seeds in the Table below  (values in parentheses represent standard error).
>
> |            | **n=40**       | **n=60**       |
> |------------|----------------|----------------|
> |            | CIG↓           | CIG↓           |
> | HCO (Ours) | 0.033 (±0.001) | 0.033 (±0.002) |
> | TAM        | 0.090 (±0.007) | 0.332 (±0.026) |
> | Flat       | 0.065 (±0.012) | 0.086 (±0.014) |
> |            | **n=80**       | **n=100**      |
> |            | CIG↓           | CIG↓           |
> | HCO (Ours) | 0.041 (±0.003) | 0.059 (±0.006) |
> | TAM        | 1.017 (±0.113) | 2.910 (±0.126) |
> | Flat       | 0.120 (±0.014) | 0.260 (±0.056) |
>
> Table. The common instance gap performance of the compared methods in generalization setting with instance size $n=40, 60, 80, 100$
>
> Our experiment result demonstrate HCO's superior performance over baseline methods across all tested instance sizes ($n=40, 60, 80, 100$), with a notably wider performance margin against Flat in larger or more novel instances ($n=80$ and $100$). This underscores the bi-level formulation's effectiveness in enhancing generalization capabilities, even in more challenging scenarios.
>
>
> * Data efficiency performance of the algorithm is similar to, rather than improving on, baselines.
>
> → Figure 3 in the paper demonstrates HCO's superior learning efficiency over baseline methods. HCO rapidly achieves an FSR of 0.7 within just 0.5 million episodes—a tenfold increase in data efficiency compared to Flat, which reaches the same FSR at 5 million episodes, but with a larger (or worse) Gap. Furthermore, HCO attains an FSR comparable to TAM's but with a Gap that is less than 1/50th of TAM's (0.025 vs 1.3 at 5 million episodes). These results confirm HCO's outstanding data efficiency relative to all baseline methods.
>
>
> **Minor**
>
> * Grammar and usage errors should be corrected in the "Experiments" section.
>
> → Thanks, we will fix the sentences.

---

### Author Response · Authors · 2024-01-29
**Common response # 1**

**1. GNN architecture design (reviewer MeyQ, MbwE)**

→ We'll first outline why we chose GAT + MHA for our architecture. GAT was selected based on its superior performance in a previous study [1] on the Steiner tree problem (STP) using reinforcement learning, outperforming other GNN architectures. For the Steiner tree packing problem (STPP), which differs from STP by having multiple terminal node types, understanding the global graph structure is essential. Therefore, we integrated a Multihead Attention (MHA) layer with GAT, influenced by research [2], [3] demonstrating MHA's effectiveness in GNNs for handling global information.

We also empirically compared our GAT+MHA with a simpler Graph Isomorphism Networks with edge (GINE) [4]. We compared the performance of our method ('Ours'), 'Flat', and 'TAM' on graphs with 40 nodes. The results of this comparative analysis are as follows.

Table 1. Comparing GNN architectures.
|            | GAT+MHA |       |  GINE |       |
|:----------:|:-------:|:-----:|:-----:|:-----:|
|            |   Gap↓  |  FSR↑ |  Gap↓ |  FSR↑ |
| HCO (Ours) |  0.040  | 0.971 | 0.039 | 0.969 |
|     TAM    |  0.107  | 0.864 | 0.957 | 0.842 |
|    Flat    |  0.072  | 0.954 | 0.045 | 0.957 |



In the case of HCO and Flat, there is no significant deviation, but in the case of TAM, we confirmed that the performance was significantly reduced in terms of gap. We believe that this result is due to the processing effect of global information due to the presence or absence of multihead attention.


[1] Du, H., Yan, Z., Xiang, Q. and Zhan, Q., 2021. Vulcan: Solving the Steiner tree problem with graph neural networks and deep reinforcement learning. arXiv preprint arXiv:2111.10810.

[2] Wu, Z., Jain, P., Wright, M., Mirhoseini, A., Gonzalez, J.E. and Stoica, I., 2021. Representing long-range context for graph neural networks with global attention. Advances in Neural Information Processing Systems, 34, pp.13266-13279.

[3] Louis, S.Y., Zhao, Y., Nasiri, A., Wang, X., Song, Y., Liu, F. and Hu, J., 2020. Graph convolutional neural networks with global attention for improved materials property prediction. Physical Chemistry Chemical Physics, 22(32), pp.18141-18148.

[4] Hu, W., Liu, B., Gomes, J., Zitnik, M., Liang, P., Pande, V. and Leskovec, J., 2019, September. Strategies for Pre-training Graph Neural Networks. In International Conference on Learning Representations.

---

### Author Response · Authors · 2024-01-29
**Common response # 2**

**2.Experiments with multiple seeds (reviewer MeyQ, yvFN)**

→ Due to computational limitations, we conducted our experiments with 4 seeds, and the result is summarized in Table 2 below. Compared to the previous result with 1 seed, the new result in Table 2 shows consistent findings with minimal variance (values in parentheses represent standard error). This consistency and small variance indicate that expanding to 6 seeds would yield comparable outcomes. Also, Table 2 demonstrates the robustness of our initial conclusions: our HCO framework significantly boosts sample efficiency and outperforms specialized heuristics (PathFinder), mathematical optimization(MILP), and tailored learning-driven algorithms for the Steiner Tree Packing Problem (STPP). We will update our paper with the new result.

Table 2. Main result table for STPP with mean and standard deviation

|     $n=40$    |                |                |          |     $n=60$    |                |                |          |
|:-------------:|----------------|----------------|----------|:-------------:|----------------|----------------|----------|
|               | Gap↓           | FSR↑           | ET (ms)↓ |               | Gap↓           | FSR↑           | ET (ms)↓ |
| HCO (Ours)    | 0.040 (±0.001) | 0.971 (±0.004) | 38       | HCO (Ours)    | 0.034 (±0.001) | 0.960 (±0.003) | 99       |
| TAM           | 0.107 (±0.008) | 0.864 (±0.012) | 17       | TAM           | 0.118 (±0.009) | 0.912 (±0.003) | 46       |
| Flat          | 0.072 (±0.011) | 0.954 (±0.07)  | 106      | Flat          | 0.064 (±0.008) | 0.927 (±0.004) | 252      |
| MILP-1s       | 0.000          | 1.000          | 127      | MILP-1s       | 0.000          | 0.982          | 501      |
| PathFinder-SP | 0.112          | 0.974          | 5        | PathFinder-SP | 0.165          | 0.990          | 8        |
| PathFinder-TA | 0.116          | 0.966          | 19       | PathFinder-TA | 0.147          | 0.974          | 48       |
| MILP-∞        | 0.000          | 1.000          | 125      | MILP-∞        | 0.000          | 1.000          | 532      |
|     $n=80$    |                |                |          |    $n=100$    |                |                |          |
|               | Gap↓           | FSR↑           | ET (ms)↓ |               | Gap↓           | FSR↑           | ET (ms)↓ |
| HCO (Ours)    | 0.054 (±0.002) | 0.931 (±0.003) | 124      | HCO (Ours)    | 0.057 (±0.003) | 0.897 (±0.009) | 246      |
| TAM           | 0.401 (±0.077) | 0.625 (±0.006) | 47       | TAM           | 1.642 (±0.550) | 0.510 (±0.035) | 47       |
| Flat          | 0.079 (±0.004) | 0.882 (±0.018) | 529      | Flat          | 0.070 (±0.013) | 0.896 (±0.007) | 679      |
| MILP-1s       | 0.001          | 0.832          | 975      | MILP-1s       | 0.000          | 0.035          | 1007     |
| PathFinder-SP | 0.150          | 0.976          | 20       | PathFinder-SP | 0.155          | 0.970          | 35       |
| PathFinder-TA | 0.149          | 0.965          | 115      | PathFinder-TA | 0.150          | 0.954          | 170      |
| MILP-∞        | 0.000          | 1.000          | 1648     | MILP-∞        | 0.000          | 1.000          | 4685     |

---

### Author Response · Authors · 2024-01-29
**Common response # 3**

**3. Broader applicability of HCO, not just for STPP (reviewer MeyQ, yvFN)**

→ While we have showcased STPP as an example of a feasibility-hard CO problem in our research, our method can be applied to other CO problems. To the best of our knowledge, there is no existing RL environment implementation of any feasibility-hard CO problems. Also, implementing the RL environment for a feasibility-hard CO problem, generating instances, and training/evaluating all the compared algorithms would currently require significant resources and time, which we may not be able to commit within the short rebuttal period. Instead, we provide below how our bi-level framework can be formulated for various CO problems to show that our method can be applied to a wide range of CO problems. Specifically, we provide the bi-level formulation for Maximal Independent Set (MIS), Set Cover (SC), and Vehicle Routing Problem (VRP), which are some of the representative classes of combinatorial optimization problems in the literature.


### Maximal Independent Set

Given a graph $G = (V,E)$, an independent set (of vertices) is a subset $S \subset V$ of vertices such that no two vertices in the set $S$ are adjacent. The maximal independent set (MIS) is the largest independent set in the graph. Formally, the problem of finding the maximal independent set can be formulated as follows:

$$
\begin{aligned}
\max_{S \subseteq V} &\quad \|S\| \\\\
\text{s.t.} &\quad (u, v) \notin E \quad \forall u, v \in S
\end{aligned}
$$

One of the possible bi-level formulations for MIS is to decompose the original problem into distinct sub-MIS problems. Intuitively, we can partition the vertices $V$ into disjoint clusters, where one can solve a smaller MIS on each of the partitioned clusters. For example, the high-level problem of a MIS can be written as follows:

$$
\begin{aligned}
\max_{y_{1}, \cdots, y_{N} \subset V} &\quad \sum_{n \leq N}\|L(y_{n})\|\\\\
\text{s.t.} &\quad y_{i} \cap y_{j} = \emptyset \quad \forall i, j \leq N \\\\
&\quad (u, v) \notin E \quad \forall u \in L(y_{i}), \ \forall v \in L(y_{j})
\end{aligned}
$$

where $y_{1}, \cdots, y_{N}$ denote the disjoint subsets (clusters) of vertices, and $L(y_{n})$ is a solution of the low-level subproblem:

$$
\begin{aligned}
L(y_{n}) := \arg\max_{S \subset y_{n}} \\\\
{ \|S\|: (u, v) \notin E, \ \forall u, v \in S \}
\end{aligned}
$$


which is simply the original MIS formulation. This formulation gets much simpler when solved with our sequential decision-making problem via an MDP formulation provided in Appendix A.1. For each step $n \leq N$, we choose a subset $y_{n} \subset V$, solve for the low-level problem $L(y_{n})$, and remove all nodes in $y_{n}$ and the nodes adjacent to ones in $L(y_{n})$. Then in the next step, we can again solve for the high-level problem in the reduced graph, just as we have done in our STPP formulation.

### Set Cover
Given a graph $G = (V, E)$, a set cover problem (SC) is to find the smallest subset $S \subseteq V$ that includes at least one endpoint of every edge of the graph. Formally, we can write:

$$
\begin{aligned}
\min_{S \subset V} &\quad \|S\| \\\\
\text{s.t.} &\quad u \in S \lor v \in S \quad \forall (u, v) \in E
\end{aligned}
$$

Similarly, as done in MIS, we can decompose the original SC instance into smaller sub-SC instances. For each high-level step $n \leq N$, we choose a subset $y_{n} \subset V$, solve for the low-level problem $L(y_{n})$ (which is a smaller SC), and remove all nodes in $y_{n}$ and the nodes adjacent to ones in $L(y_{n})$.

### Vehicle Routing Problem

The vehicle routing problem (VRP) is finding the best way for a number of vehicles to deliver things in a route using the least time or distance. The VRP is arguably similar to the STPP in nature: If the goal of an STPP instance is to find a packing of disjoint Steiner trees, the objective in a VRP instance is to find a packing of disjoint traveling salesman problem (TSP) routes. Hence, the bi-level formulation of VRP is straight-forward: For each step of the high-level step $n \leq N$, we find a subset $y_{n} \subset V$ of customers that a $n$-th vehicle is to visit. Then, the corresponding low-level problem is to find a TSP solution within the subset of nodes $y_{n}$.

As such, HCO can be broadly applicable to most general CO problems

---

### Decision · Action_Editor_JvnB · 2024-03-05

**Recommendation:** Reject

**Comment:**

In general, responses made by the authors were good, clarifying some of concerns.
However, Reviewer MbwE claimed that his/her several major concerns were not adequately addressed in the author response.
In particular, an ablation of the behavior cloning step was not provided in the time allowed.
I am sure that the authors will be able to do this if more time is permitted.
I would like to suggest that the authors may consider submitting a major revision at a later time.

**Audience:**

The proposed algorithm is well-suited to the structure of the Steiner Tree Packing Problem.
However, the decomposition seems somewhat problem specific, so its broad applicability is very limited.
It won't be easy for someone to use this algorithm to solve other combinatorial optimization problem with similar constraints.

**Claims And Evidence:**

This paper presents a ML-based algorithm to tackle feasibility-hard combinatorial optimization problems. The method decomposes the combinatorial optimization problem into a bilevel problem so that solutions to smaller problems are combined into the final solution. The paper focused on the Steiner Tree Packing Problem (STPP) to justify the validity of the proposed method. In general, the paper is well written.
The algorithm design is clear and well-suited to the structure of the Steiner Tree Packing Problem.
Experiments do not support the claim sufficiently. In particular, an ablation of the behavior cloning step was not provided in time for the response. Thus, it is hard to assess which of the gains come from behavior cloning vs. the actual algorithm proposed. In addition, while the proposed algorithm does well in optimality and feasibility, the baselines also do well, making it difficult to assess the gains from this particular algorithm. As such, there is a question as to whether the evidence in the experiments sufficiently supports the claims of the paper.

**Resubmission Of Major Revision:**

The authors may consider submitting a major revision at a later time.